# Nanomaterials-Based Wound Dressing for Advanced Management of Infected Wound

**DOI:** 10.3390/antibiotics12020351

**Published:** 2023-02-08

**Authors:** Qian Pang, Zilian Jiang, Kaihao Wu, Ruixia Hou, Yabin Zhu

**Affiliations:** Department of Cell Biology and Regenerative Medicine, School of Medicine, Ningbo University, Ningbo 315211, China

**Keywords:** nanomaterials, antimicrobial, bacterial infection, infected wound, wound healing

## Abstract

The effective prevention and treatment of bacterial infections is imperative to wound repair and the improvement of patient outcomes. In recent years, nanomaterials have been extensively applied in infection control and wound healing due to their special physiochemical and biological properties. Incorporating antibacterial nanomaterials into wound dressing has been associated with improved biosafety and enhanced treatment outcomes compared to naked nanomaterials. In this review, we discuss progress in the application of nanomaterial-based wound dressings for advanced management of infected wounds. Focus is given to antibacterial therapy as well as the all-in-one detection and treatment of bacterial infections. Notably, we highlight progress in the use of nanoparticles with intrinsic antibacterial performances, such as metals and metal oxide nanoparticles that are capable of killing bacteria and reducing the drug-resistance of bacteria through multiple antimicrobial mechanisms. In addition, we discuss nanomaterials that have been proven to be ideal drug carriers for the delivery and release of antimicrobials either in passive or in stimuli-responsive manners. Focus is given to nanomaterials with the ability to kill bacteria based on the photo-triggered heat (photothermal therapy) or ROS (photodynamic therapy), due to their unparalleled advantages in infection control. Moreover, we highlight examples of intelligent nanomaterial-based wound dressings that can detect bacterial infections in-situ while providing timely antibacterial therapy for enhanced management of infected wounds. Finally, we highlight challenges associated with the current nanomaterial-based wound dressings and provide further perspectives for future improvement of wound healing.

## 1. Introduction

Every year, hundreds of millions of people around the world suffer from external (e.g., abrasions, surgery and burns) or endogenous (e.g., diabetes and blood vessels diseases) skin wounds. Studies have shown that the annual cost of treating these wounds is $25 billion in the United States alone [1,2]. These heavy health and cost burdens have therefore made wound healing a center of research focus. Normal wound healing is a complex and ordered biological process, and can be divided into the following four classical and overlapping stages: hemostasis, inflammation, proliferation and remodeling [3,4]. The Hemostasis stage usually starts as soon as the injury occurs, in which the activation of platelets contribute to the formation of a fibrin clot to be used as a hemostatic plug to minimize hemorrhage. In inflammatory stage, neutrophils and macrophages are recruited to the wound site to clear dead cells and microbes and protect the wound from infection. In addition, these inflammatory cells also induce the initiation of proliferation phase by secreting growth factors and cytokines. The proliferation stage is the phase of angiogenesis, granulation tissue formation, and re-epithelialization, which arises about 2–10 days after the injury. The remodeling stage begins 2–3 weeks after injury, in which scar tissues are formed through reorganization and rearrangement of collagen fibers. For acute wounds, defects can usually be healed in an orderly manner within 2–3 weeks. However, the healing process may be easily disturbed by internal and external factors, such as diabetes, vascular disease, poor nutrition, and microbial infections, degeneration of wounds into chronic and non-healing subtypes [5,6]. Notably, wound infections caused by the colonization of bacteria and other microorganisms at the wound site poses a major challenge to wound management, due to the induction of severe inflammatory reactions that delay wound healing and even cause wound deterioration [7,8,9]. For diabetes patients, diabetic foot ulcers represent severe chronic wounds as they are easily infected by bacteria due to the high glucose level at the wound site [10]. Previous estimates show that a diabetic patient has a 25% chance to develop a foot ulcer [11]. Studies have shown that non-healed bacterial-induced diabetic foot ulcers account for 0.21−1.37% of all diabetic patients, with a five-year mortality rate of 80% [11,12]. Most of these cases require limb amputation. Therefore, a key goal of wound management entails the prevention and treatment of bacterial infections, with a view of promoting wound repair, improving healing outcomes, and saving patients’ lives.

Wound dressings have long predisposed wounds to bacterial invasion. Studies have shown that traditional wound dressings, such as use of gauze and cottons, as well as modern techniques such as hydrogels, hydrocolloid, foams, and polymeric films can promote wound healing [13,14,15]. However, bioactive wound dressings that integrate antibiotics or other antibacterial components are not only more effective at combating bacterial infections but also accelerating the wound repair process than the above-mentioned passive wound dressings [8,16]. Traditional antibiotics, such as tetracyclines, aminoglycosides, quinolones, and cephalosporins, have proved effective in killing various bacteria by destroying the bacterial cell wall due to their effect on protein and nucleic acid synthesis [17]. However, the misuse and abuse of antibiotics has resulted in development of multiple drug-resistant bacteria and formation of biofilms [8,18]. Studies have shown that *Staphylococcus aureus*, methicillin-resistant *S. aureus* (*MRSA*), and *Pseudomonas aeruginosa*, which commonly infect wounds, exhibit differential drug resistance to more than one kind of antibiotics [19,20]. Moreover, biofilms produced by bacteria have been found to prevent antibiotic penetration thereby exacerbating drug resistance, and constraining wound infection [21,22]. Therefore, the rapid increase in the incidence of wound infections, coupled with the severe consequences such as limb amputation deaths associated with chronic wounds and the dangers of antibiotic strategies for wound infection, necessitates the urgent development of novel technologies and platforms to improve treatment efficacy and accelerate healing of wound infections.

Nanomaterials, with a diameter less than 100 nm, have been widely applied to control bacterial infections and promote wound healing due to their unique small size and surface effects, as well as tunable physiochemical properties [23,24,25]. For example, metal (silver, gold, copper, and zinc oxide) nanoparticles with intrinsic antibacterial properties, have been widely used to not only improve antibacterial efficiency but also reduce bacterial drug resistance due to their high surface-to-volume ratio attributes that allow for enough contact and interaction with bacteria [26]. In addition, the tunable shape and functional modification of nanomaterials also endow them with ideal capability for delivery of antibacterial drugs or bioactive molecules to wound sites thus promoting treatment efficacy [27]. Furthermore, special optical properties in some of the nanomaterials, such as those made of metal oxides, carbon- and polymer-based nanomaterials, have allowed them to be efficient photothermal therapy (PTT) and photodynamic therapy (PDT) agents to kill bacteria through local heating or controlled ROS productions in the infected wound without the trouble of bacteria resistance [28]. Collectively, these evidences indicate that nanomaterials-based approaches are superior to traditional antibacterial strategies for wound healing applications [29,30]. However, the application of nanomaterials alone at the wound site is constrained by safety and efficacy challenges [31]. Therefore, incorporating nanomaterials into existing wound dressings to develop nanomaterials-based wound dressing platforms may be a promising strategy for safer and more efficacious management of infected wounds.

In recent decades, various nanomaterials with intrinsic antibacterial performances and reservoir capacity of therapeutic agents have been widely developed and incorporated into wound dressing for wound repair applications [29,30,32,33]. Notably, early detection of bacteria is better than antibacterial treatment strategies, especially for the management of chronic wounds, since it offers an opportunity for the timely diagnosis of infections to guide the development of appropriate and efficacious treatment strategies [34,35]. Technological advancements in the nano-fabrication field has led to the development and wide application of functional nanomaterials in biosensing and disease diagnostics [36,37]. However, the abovementioned articles mainly focused on use of nanomaterials to control infections, with little information on their application in detection and treatment of bacterial infections. In this review, we first summarize recent research progress on the use of nanomaterials-based wound dressing to control infections via the integration of nanoparticles with intrinsic antibacterial performances (metals and metal oxide nanoparticles). Next, we describe application of nanomaterials-based antibacterial wound healing systems using nanomaterials (such as polymer, lipid and inorganic nanoparticles), as passive drug carriers to deliver and release antibacterial agents in a sustained manner or as stimuli-responsive drug carriers to release antibacterial agents in a controllable manner triggered by endogenous bacterial microenvironment or external stimuli. Focus is given to nanomaterials with the ability to kill bacteria by photo-triggered heat (PTT) or ROS (PDT). Finally, we highlight state-of-the-art nanomaterials-based wound dressings that confer all-in-one bacterial detection and infection control capacity, and further discuss the associated challenges and future perspectives.

## 2. Integrating Antibacterial Nanomaterials into Wound Dressings for Infection Control

The unique physicochemical properties of nanomaterials, including high surface to volume ratios, adjustable sizes, shapes and surface chemistry, has endowed them with a novel capability to combat bacteria, especially multidrug resistant (MDR) species and biofilms. Nanoparticles with intrinsic antibacterial characteristics, such as metallic and metallic oxides, represent a major class of nanomaterials used in wound healing. In addition, nanomaterials have been used as effective carriers to store and deliver antibacterial agents to wound sites. Integrating these nanomaterials into hydrogel-based wound dressings has been shown to promote antibacterial activity against bacteria, thus making them ideal candidates for wound repair.

### 2.1. Metal and Metal Oxide Nanoparticles as Intrinsic Antibacterial Treatment Agents

Metal NPs, such as silver nanoparticles (AgNPs), gold nanoparticles (AuNPs), copper nanoparticles (CuNPs) and zinc oxide nanoparticles (ZnO NPs), have been widely applied as intrinsic antibacterial agents. These agents have exhibited excellent bactericidal activity against both MDR bacteria and biofilms [38,39]. It has been reported that multiple mechanisms are responsible for the excellent antibacterial performances of Metal NPs [26,38,39,40]. The possible antibacterial mechanisms are illustrated in detail in Figure 1 in the previous reported work [41]. For instance, metal NPs pierce the cell membrane with their sharp edges, thereby improving permeability of the bacterial cell membrane. They also interact with bacterial membrane proteins, thereby causing the leakage of intracellular components and lysis of bacteria. Moreover, metal NPs or metallic ions released from them destroy the structure of proteins, DNA, and lipids by binding via sulfur bonds or inducing oxidative stress, thereby causing bacterial death.

**Silver nanoparticles (AgNPs)**. AgNPs have been widely applied as a broad-spectrum antimicrobial agent for treatment of burns and ulcers wounds for the infection control. Studies have shown that these agents accelerate wound repair by killing bacteria, destroying formation of biofilms, suppressing inflammation, and improving tissue re-epithelization [42,43,44]. AgNPs’ antibacterial activity is attributed to a release of silver ions from oxidized AgNPs under acidic conditions, which subsequently kill bacteria via the destruction of cell walls, DNA, Adenosine triphosphate (ATP), and the production of reactive oxygen (ROS) [45]. Studies have shown that AgNPs’ antibacterial activity exhibits a size-dependent trend, with a decrease in the size of AgNPs associated with an elevated release of silver ions which subsequently promote antimicrobial activity [46,47]. Considering their excellent antibacterial activity, AgNPs can be loaded into various wound dressings for the treatment of wound infections. Some commercially available AgNPs-coated wound dressings, such as “Acticoat” have been successfully used in clinical applications, and have been found to confer sustained release of silver ions thereby suppressing bacterial infections and promoting wound healing [48]. However, AgNPs-based wound dressing therapies are constrained by toxicity concerns resulting from the uncontrolled release of silver ions. Notably, some studies have shown that high AgNPs concentrations not only cause a dose-dependent DNA damage but also suppress the viability of keratinocyte and metabolism cells [44,49]. To reduce these adverse effects, a low AgNPs concentration in combination with antibiotics is recommended. A previous study demonstrated that AgNPs and tetracycline synergistically interact for enhanced antibacterial effects [50]. Additional approaches for improving antibacterial properties and minimizing the cytotoxic effects of AgNPs include coating them with biomolecules as well as incorporating them into hydrogel-based wound dressings [51,52,53,54]. Wu et al. and Pal et al. described in-situ synthesis of AgNPs on bacterial cellulose-based porous hydrogel network. Results showed that the developed wound dressings not only had high bacteria-killing activity but were also associated with low toxic events [52,53]. In another study, incorporation of collagen-coated AgNPs into collagen hydrogels resulted in efficacious and safe application of AgNPs [54]. Notably, the resulting hybrid wound healing not only retained the biocompatibility for human skin fibroblasts and keratinocytes but also exhibited remarkable antimicrobial activity against *S. aureus*, *S. epidermidis*, *E. coli* and *P. aeruginosa*. In addition, combing AgNPs with chitosan, polyvinyl alcohol (PVA), lignin and zwitterionic hydrogels successfully reduced silver cytotoxicity and improved infection control [55,56,57,58]. Researchers recently developed novel silver-lignin nanoparticles (Ag@Lig NPs) crosslinked thiolated hyaluronic acid (HA-SH) based hybrid nanocomposite hydrogels for chronic wound management, with lignin used as a reducing and capping agent of AgNPs (Figure 2) [59]. Notably, the Ag@Lig NPs-loaded hydrogel exhibited strong antibacterial activity against both Gram-negative and Gram-positive bacteria, with no toxicity to human keratinocytes after 7 days of direct contact. In vivo results showed that it effectively suppressed wound inflammation and promoted tissue remodeling in a diabetic mouse wound model, affirming its potential for the management of chronic wounds. In addition to hydrogel, electrospun nanofibers are also ideal carriers for Ag NPs nanoparticles and present promising applications for wound healing [60,61]. For example, Liu et al. developed a curcumin@β-CD/AgNPs nanoparticle and loaded the composite nanoparticle into a chitosan electrospun nanofiber [61]. The developed functionalized silver nanoparticles showed less toxicity and effective antibacterial activity against *P. aeruginosa*, *S. aureus* and *E. coli*. In vivo, curcumin@β-CD/AgNPs-loaded chitosan dressing showed more uniform collagen distribution, accelerated wound closure rates, and less scar formation compared to commercial AgNPs dressing.

**Gold nanoparticles (AuNPs).** Apart from AgNPs, AuNPs have also been extensively applied in infection therapy, wound healing, and tissue regeneration due to their excellent biocompatibility, good optical and chemical stability as well as ease of surface modification [42,62,63,64]. AuNPs differ from AgNPs in that they release Ag^+^ which possess good bacteria-killing capability by directly targeting and destroying their cell walls to cause loss of intracellular components, inhibit ATP synthesis and interfere with DNA transcription. Studies have demonstrated that AuNPs exert broad-spectrum antibacterial activity against both Gram-negative and Gram-positive bacteria, including *E. coli*, *P. aeruginosa*, *S. aureus*, and *MRSA* [63,64]. AuNPs can also efficiently absorb near infrared light and convert it into heat, which further endows them with the photothermal bactericidal ability [65]. Moreover, AuNPs act as antioxidants and have been shown to promote wound repair by regulating anti-inflammatory cytokines, suppressing ROS levels and promoting angiogenesis [66,67]. However, using AuNPs alone at the wound site is constrained by rapid loss and the associated toxic side effects. Consequently, researchers have integrated AuNPs into a wide range of hydrogel wound dressings with the aim of improving their efficacy and reducing the associated side effect [68,69]. For example, AuNPs with different shapes (nanorods and nanospheres) and surface modifications (neutral, positive and negative charged polymers) were prepared and embedded into a thermosensitive hydrogel of poloxamer 407 (Figure 3) [69]. The resulting hydrogel exhibited the slow and prolonged release of AuNPs, thus enhancing the topical application of AuNPs at the wound site. Both neutral and positively charged polymers modified AuNPs-based hydrogels and exhibited excellent antibacterial activity against *S. aureus* and *P. aeruginosa,* in vitro and accelerated skin re-epithelization and collagen deposition in an in vivo animal model. Notably, low percentages of AuNPs were deposited on the main animal organs after 21 days of wound treatment, indicating that the combination of AuNPs with hydrogel minimized the side effects of AuNPs.

**Zinc oxide nanoparticles (ZnO NPs)**. Just like AgNPs, ZnO NPs are also considered an excellent antibacterial agent, although their antibacterial activity depends on release of free Zn^2+^ from ZnO [70,71]. Numerous studies have shown that the antibacterial effect of ZnO NPs is strongly correlated with the size and concentrations of the nanoparticles [72,73]. For example, 8-nm ZnO NPs were found to kill 95% of *S. aureus* at a low concentration (1 mM) but those with a larger size (50−70 nm) only caused 40−50% sterilization at a high concentration (5 mM) [73]. Although the Food and Drug Administration (FDA) has confirmed safety and consequently approved most ZnO materials, studies have shown that high concentrations of nano-sized ZnOs exhibit obvious toxicity to normal cells, such as keratinocytes, by inducing mitochondrial dysfunction and causing cell membrane oxidative stress and apoptosis, which greatly limits application of ZnO NPs for wound healing treatment [23]. To solve this problem, Manuja et al. [71] loaded ZnO NPs into sodium alginate-gum acacia hydrogels (SAGA) and found a sustained release of ZnO NPs at lower doses. More importantly, the antibacterial and healing effect was retained while the toxic effects were low [74]. In addition, ZnO NPs have also been combined with other natural polysaccharides polymers-based electrospinning nanofibers such as chitosan and polyvinyl alcohol, and the resulting nanocomposite membranes demonstrated improved antibacterial and wound healing effects [75]. Overall, these reports indicate that ZnO NPs/hydrogel-based platforms hold great promise for wound healing applications.

**Bimetallic nanoparticles**. Considering the bactericidal capacity of various metal NPs, researchers prepared bimetallic NPs (by combining two types of metals) and found that these remarkably improved bactericidal efficiency [76,77,78]. For instance, Li and coworkers synthesized bimetallic Au-Ag NPs and loaded them into a chitosan hydrogel. Results showed that the novel agent had significantly higher antibacterial activity against both *S. aureus* and *E. coli* than monometallic-loaded chitosan hydrogel (Figure 4) [79]. Apart from remarkably promoting wound healing effects, this wound dressing was not nontoxic to L929 cells. Similarly, Khan et al. covered biogenic AuNPs with a thin layer of ZnO and prepared the Au@ZnO core-shell nanocomposites. Their results revealed that these nanocomposites remarkably promoted antibacterial and anti-biofilm activity against *S. aureus* and methicillin resistant *Staphylococcus haemolyticus* (*MRSH*) without provoking nuclear damage and cytotoxicity to mouse fibroblasts cells [80].

### 2.2. Nanomaterials as Passive Carriers of Antibacterial Agents for Antibacterial Treatment

Apart from antibacterial activities, nanomaterials can also be used as effective drug delivery systems to facilitate controlled or sustained release of antibacterial agents during treatment of bacterial infections. For example, studies have shown that nanocarrier-based drug delivery systems can improve the bioavailability of poorly water-soluble drugs, prolong drug half-life, optimize the pharmacokinetics profiles, and reduce the frequency of drug administration, thereby remarkably promoting the therapeutic effect [81]. Incorporating these nanocarriers into wound dressings has emerged as an effective strategy for treatment of infected wounds. One of the most widely applied nanocarriers-based drug delivery systems is nanomaterials, which are usually fabricated from polymers, lipid-based and inorganic materials. These systems remarkably release antibacterial agents in a sustained and passive manner [82].

**Polymer-based nanocarriers for antibacterial therapy.** Polymer-based nanocarriers are usually synthesized from various natural or synthetic polymers through different assemble methods that are based on electrostatic interactions, hydrophilic and hydrophobic interactions or host-guest interactions. Due to their good biocompatibility, hydrophilicity, and versatility that can be tailored to satisfy the needs of delivered drugs, they are ideal drug delivery systems for wound healing applications [83,84]. For instance, natural polymers such as chitosan (CS) and synthetic polymers including polycaprolactone (PCL) and poly (lactic-co-glycolic acid) (PLGA) have been widely developed as nanocarriers for the delivery of antibacterial agents [85,86,87]. Chitosan and its derivatives are cationic polymers with intrinsic antibacterial capacities that can effectively promote the wound healing process by improving inflammatory cell functions at the wound site, stimulating fibroblasts and osteoblasts [88]. Chitosan NPs can be formed via different methods, including micro-emulsion, ionotropic gelation and emulsification-solvent diffusion as well as evaporation [85]. Antibacterial agents such as antibiotics, antibacterial peptides or small molecule drugs are loaded into CS NPs through hydrophobic hydrogen bonding or electrostatic interactions via direct encapsulation during the preparation process or soaking after CS NPs formation. In a previous study, curcumin, an antimicrobial and antioxidant agent with poor water solubility and impaired skin penetration capacities, was loaded into CS NPs and embedded in fiber-based wound dressings [89]. The dressing allowed for the gradual release and enhanced the wound healing effects of curcumin. Due to their versatile degradation kinetics and controlled drug release properties, PLGA NPs have also been widely used as drug carriers [90]. The PLGA NPs that are prepared by water/oil/water (W/O/W) double emulsion method exhibit a high drug encapsulation efficiency and relatively uniform size distribution [91]. The PLGA NPs have a 70% encapsulation efficiency of LL-37 antimicrobial peptides and are effective treatment options for polymicrobial infected wounds [91,92]. Studies have investigated the significance of NO in antibacterial and wound-healing applications, particularly in combating MRSA biofilms [93]. However, the short half-life and limited diffusion distance of NO limits its direct applications at wound sites. Hasan et al. developed an NO delivery system by doping an NO donor of polyethylenimine/diazeniumdiolate (PEI/NONOate) with PLGA NPs [94]. The PLGA-PEI/NO NPs exhibited a sustained release of NO for over 4 days and good anti-biofilm activities, resulting in accelerated healing of MRSA biofilm-infected wounds in diabetic mice. Nanoparticles that are synthesized by natural or synthetic polymers with cationic charges are also effective antibacterial agents and carriers of anionic drugs that can kill bacteria when combined with the negatively charged bacterial cell walls. In a previous study, a cationic polymer (polyethylenimine (PEI) modified clindamycin-loaded PLGA NPs (Cly/PPNPs)) bound MRSA surfaces and exhibited enhanced antibacterial abilities as well as wound re-epithelialization when compared to NPs without PEI [95]. Liu et al. fabricated a self-assembled nanofiber using the cationic poly (ethylene teraphthalate) (PET) and used it as a carrier for the delivery of anionic antibiotics, piperacillin-tazobactam (PT) [96]. Despite the advances in cationic polymeric nanoparticles in infection control, the potential cytotoxic effects of cationic polymers to normal cells and tissues have yet to be conclusively determined. Embedding these NPs into hydrophilic hydrogel networks or modifying them with hydrophilic polymers such as polyethylene glycol (PEG) and sorbitol are effective strategies for suppressing the cytotoxic effects of cationic polymer nanoparticles without affecting their therapeutic efficiency [97,98,99,100].

**Lipid-based nanocarriers for antibacterial therapy**. Because of their small particle sizes and good compatibility, lipid-based nanoparticles are attractive carriers of antibacterial drugs for treatment of infected wounds. Liposomes have been investigated as lipid-based nanoparticles. They are fabricated by a hydrophilic inner core and a hydrophobic shell and have good capacities for encapsulating both hydrosoluble and liposoluble antibacterial agents [101]. Liposomes can facilitate the penetration of antibacterial agents into bacterial cells, enhancing the bactericidal effects [102,103]. Various antibiotics have been loaded into different liposomes and successfully inhibited bacterial infection [102,103,104,105]. However, topical use of liposomes at wound sites is often limited by their low drug encapsulation efficiency (less than 30%) and rapid drug release [106]. To improve their efficacies, surface modification and hydrogel encapsulations of liposomes can increase the drug loading level and prolong drug release in wound sites. Thapa et al. designed collagen mimetic peptide (CMP)-modified vancomycin-loaded liposomes (CMP-Van-Lipo) and embedded the liposomes into a collagen hydrogel [107]. Vancomycin was continuously released from the wound dressing for 48 h in vitro and enhanced the antibacterial effects of wound dressing against MRSA in vitro and in vivo. Similar to liposomes, colloidal layered vesicles that are usually composed of non-ionic surfactants and amphipathic compounds (called niosomes) are also ideal carriers for the delivery of both hydrophobic and hydrophilic antibacterial agents [108,109]. Compared to liposomes, niosomes are more stable, affordable and allow better loading and higher permeability for small molecules and ions [106]. Abdelaziz et al. reported that the norfloxacin-loaded niosomes proved excellent bactericidal ability towards *Pseudomonas aeruginosa* biofilm [110]. Encapsulating electrosprayed cefazolin-loaded niosomes onto electrospun chitosan nanofibrous membrane also demonstrated effective antibacterial property against both Gram-negative and Gram-positive bacteria, good biocompatibility and enhanced skin regeneration by improving re-epithelialization, tissue remodeling, and angiogenesis [111]. Apart from liposomes and niosomes, lipid nanoparticles, including solid lipid NPs (SLNs) and nanostructured lipid carriers (NLCs) are effective antibacterial agent carriers for infected wound treatment [112,113]. Compared with liposomes, lipids NPs exhibit increased drug encapsulation efficiencies, decreased drug leakage and sustained drug release profiles, which are attributed to their denser structures [114]. The SLNs have a drug encapsulation efficiency of between 50% and 100% while NCLs have an encapsulation efficiency of around 85% [106]. Many studies have reported increased antibacterial effects when using antibacterial agents loaded SLNs and NCLs systems [115,116,117,118,119,120]. In one study, the LL37 antimicrobial peptide was encapsulated by SLNs, and was continuously released for over a period of 14 days and effectively killed *S. aureus* and *E. coli* [116]. In another study, the peppermint essential oil-loaded NLCs exhibited high-efficient antibacterial activities against *S. epidermidis*, *S. aureus*, *E. coli* as well as *P. aeruginosa* and accelerated wound healing effects [119]. To apply SLNs and NLCs in wound healing, these NP carriers have also been incorporated into hydrogel wound dressings. Silver sulfadiazine (SSD)-loaded SLNs was embedded into chitosan hydrogels for SSD delivery [120]. The hydrogel exhibited prolonged SSD release to wound sites and higher antibacterial efficiencies against planktonic bacteria and *P. aeruginosa* biofilms.

**Inorganic nanocarriers for antibacterial therapy.** With advanced development of chemical synthesis and preparation technologies, various types of inorganic NPs, including metallic, ceramic and carbon-based NPs have emerged as attractive drug carriers for wound healing. The inorganic NPs have a high surface area, tunable size and versatility that allow the antibacterial agents to be loaded or grafted on pore surfaces of corresponding NPs [121,122]. Among the inorganic nanoparticles, metal NPs such as AuNPs and AgNPs with intrinsic antibacterial performances have the capacity for conjugating with antibacterial agents to achieve synergistic antibacterial effects [123,124,125,126,127]. For example, conjugating ceftriaxone with AgNPs and AuNPs has been proved to enhance the bactericidal ratio against *E. coli* to 2- and 6-fold, respectively than that of pure ceftriaxone [123]. Moreover, peptide-coated AuNPs [124], Rauwolfia serpentina-modified AuNPs [125], tannic acid chelated Ag nanoparticles (TA-Ag NP) [126] and Zn-penicillin complex [127] have all been shown to have enhanced antibacterial activity and potential application for the healing of infected wounds. In addition, because of their ultra-high specific surface areas, excellent biocompatibility and biodegradability, porous structures and tunable pore sizes to ensure maximum drug loading and controlled release kinetics, metal-organic frameworks (MOFs)-based nanoplatforms (e.g., Zn-based MOFs, Cu-based MOFs, Fe-based MOFs and Ag-based MOFs) are ideal candidates for delivery of antibacterial agents [128]. In a proof-of-concept study, three types of MOFs, including Cu-MOFs 1, Co-MOF 2 and Zn-MOF 3 were prepared and respectively embedded in the PEG hydrogel [129]. Hydrogel@Cu-MOF 1 and hydrogel@Co-MOF 2 exhibited excellent antibacterial activities against both Gram-negative (*E. coli*) and Gram-positive (*S. aureus*) bacteria, when compared with hydrogel@Zn-MOF 3. Hydrogel@Cu-MOF 1 exhibited 99.9% antibacterial effects at the minimum bactericidal concentration without exerting cytotoxic effects to human dermal fibroblasts. The MOFs-based platform can be used as nanozyme catalysts to induce the ROS-induced bacterial killing effects [130,131]. In a typical study, glucose oxidase (GOx) was loaded into a Cu^2+^ doped graphited imidazolate framework (ZIF-8). Then, the as-prepared Cu@ZIF/GOx was encapsulated in a bacterial cellulose (BC) reinforced guar gum (GG) hydrogel (Figure 5A) [131]. Loading of GOx has been proven to be capable of consuming glucose to produce H_2_O_2_ and glucose acid, which induces Cu@ZIF to produce high oxidative •OH to kill bacteria. The as-prepared hydrogel exerted significant antibacterial efficacies as well as inhibitory effects on biofilm formation, implying that they have promising applications in diabetic infected wound healing (Figure 5B). In tandem with MOFs-based nanocarriers, due to their porous structures and adjustable surface modifications, ceramic-based NPs, especially mesoporous silica NPs (MSN), are used as drug carriers. Until now, various antibacterial agents including gentamicin [132], ciprofloxacin hydrochloride/Zn complex [133], AgNPs [134], and silver-bismuth nanoparticles (Ag-Bi NPs) [135] have been loaded into MSNs to treat infections in wounds. Cao et al. designed mesoporous silica supported silver-bismuth (Ag-Bi@SiO_2_) NPs by loading Ag-Bi NPs into porous SiO_2_ NPs for treatment of bacterial infected wounds (Figure 5C) [135]. The synergistic antibacterial effects against both MRSA and MRSA biofilms, which are attributed to the hyperthermia produced by Bi NPs under near-infrared light (NIR) irradiation and accelerated Ag^+^ release, have been observed in vitro. A 69.5% decrease in MRSA biofilm biomass was achieved using Ag-Bi@SiO_2_ NPs at a concentration of 100 µg mL^−1^ with an NIR laser, exhibiting better bactericidal effects than Bi@SiO_2_ NPs with laser (26.8%) or Ag-Bi@SiO_2_ NPs without laser treatment (30.8%) groups (Figure 5D). The in vivo test indicated that Ag-Bi@SiO_2_ NPs were able to kill 95.4% of bacteria in infected wounds and accelerated the wound healing process (Figure 5E). In addition, bioceramics nanoparticles such as bioactive glass nanoparticles (BG NPs), which contains the oxides of silicon, sodium, calcium, and phosphorus, exhibit good biocompability and wide applications in tissue engineering [136,137]. Using structure-forming agents, mesoporous or hollow structured BG NPs can be formed and doped with ZnO or Ag to achieve an antibacterial property for infection treatment [138,139,140]. It is worth mentioning that many reserachers have demonstrated that bioactive glass has the ability of promoting angiogenesis and skin regeneration at the wound site through promoting the production of vascular endothelial growth factor (VEGF) and fibroblast growth factor 2 (FGF2) [139]. Therefore, antibacterial BG NPs exhibit unparalleled advantage for wound healing. In addition to the above, carbon-based nanomaterials, such as graphene oxide (GO), carbon nanotubes (CNTs) and carbon-based quantum dots (CDs) are also good carriers of antibacterial drugs since their abundant oxygen-containing groups, including hydroxyl, epoxy, and carboxyl groups of carbon-based nanomaterials are active sites for the conjugation of therapeutic agents [141,142]. In their study, Patarroyo et al. grew Ag NPs in situ on graphene oxide surfaces and the as-prepared GO-Ag NPs nanoconjugates were subsequently encapsulated into gelatin-polyvinyl alcohol-hyaluronic acid hydrogels. The hydrogel exhibited 100% antibacterial effects and very low hemolytic effects (less than 5%), showing the potential for wound healing [143].

### 2.3. Stimuli-Responsive Nanomaterials Carriers of Antibacterial Agents for Antibacterial Treatment

Despite advances in the above nanocarriers for antibacterial therapy, passive antibacterial mechanisms of such drug delivery systems that rely on uncontrollable and untimely release of antibiotics, antibacterial ions and peroxide radicals may greatly decrease the treatment effects of nanomaterials, increasing bacterial drug resistance and occurrence of undesired side effects. Excess release of metal ions and ROS may be cytotoxic to normal tissue cells and trigger additional inflammatory reactions to impede wound healing while leading to other diseases [144,145]. Developing smart nanomaterials-based carriers that can respond to endogenous microenvironments of bacterial infections such as acidic pH, excess ROS, specific toxins and enzymes secreted by bacteria or external stimuli, including light, thermal and magnet to release antibacterial agents or trigger their own antimicrobial properties in a controllable manner may be an effective strategy [146]. Until now, such smart nanocarriers have been widely investigated and can be further integrated into wound dressings to promote wound repair. Examples of these advanced wound dressing systems for antibacterial therapy are discussed in the following sections.

**Endogenous stimuli-responsive nanocarriers for antibacterial therapy.** Bacterial colonization at the wound site results in significant changes in physicochemical markers in their surrounding microenvironments. For instance, the lactic acid produced by bacteria decreases the pH value of the microenvironment and induces an acidic microenvironment [147,148]. Some novel pH-responsive antibacterial nanocarrier platforms have been fabricated by utilizing pH-cleavable chemical bonds (e.g., ester bond, Schiff base bonds and hydrazone bonds) or pH-ionizable groups (e.g., zwitterionic, chitosan, and MOFs), which allows the nanocarriers to be broken or change their surface charge at acidic pH values to release antibacterial agents or change from a negative status to a positive status to effectively kill bacteria [147,148,149,150,151,152]. Wang et al. developed a photothermal agent (cypate) and a procollagen component (proline) loaded pH-sensitive octapeptide (IKFQFHFD) nanofiber-based hydrogel network for treatment of chronically infected wounds (Figure 6A) [152]. At acidic environments, the octapeptide would change from electrically neutral to a positive antibacterial peptide, resulting in disassembly of the hydrogel and release of antibacterial IKFQFHFD and cypate. In vitro, the synergistic antibacterial effects of IKFQFHFD and cypate effectively alleviated a mature MRSA biofilm. The synergistic release of proline from the hydrogel promoted tissue cell regeneration. In vivo, the as-prepared nanofiber hydrogel enabled the complete repair of MRSA biofilm wound infections within 20 days, implying that they are potential chronic wound dressings. For chronic wounds, an alkaline pH value of 8–10 at the local wound was detected. Loading of chlorhexidine (CHX) into pH-sensitive silica nanoparticles (SiNPs) achieved a 4–5-fold release of CHX on alkaline wounds than at neutral as well as acidic environments and exhibited a 4-log reduction of bacterial cells, including both Gram-negative and -positive bacteria [153]. The as-prepared CHX-SiNPs were further encapsulated into alginate hydrogels, exhibiting good bactericidal capacities against *E. coil* infected chronic wounds. Apart from changes in pH values, bacteria such as *S. aureus* and *P. aeruginosa* secrete many factors, including toxins (e.g., α-toxin) and enzymes (e.g., hyaluronidase, gelatinase, and phospholipase) [154,155]. Nanocarriers that can be specially broken by these bacteria-secreted toxins and enzymes are promising strategies for inducing the release of antibacterial drugs and achieve on-demand infection treatment [156,157,158,159]. In a previous study, a gentamicin loaded mesoporous silica nanoparticle (MSN) core was modified by a bacterial toxin-cleavable lipid bilayer shell, which was further modified by a bacteria-targeting peptide ubiquicidin (UBI29-41) [157]. The final nanoparticles were capable of targeting the bacterial infection site and achieving the rapid release of gentamicin due to bacterial toxins-induced degradation of the lipid bilayer shell. In vitro and in vivo proliferation of *S. aureus* is effectively inhibited by nanoparticles. In another recent study, Mir et al. designed bacterial lipase-cleavable poly(ε-caprolactone) nanoparticles and loaded them with carvacrol (CAR) [158]. By incorporating CAP-PCL nanoparticles into a hydrogel matrix, there was significant bacterial-induced release of carvacrol and excellent antimicrobial activities against MRSA. Trusek et al. developed a novel enzyme-responsive graphene oxide (GO)-based nanocomposite hydrogel, achieving controlled antibiotic delivery (Figure 6B) [159]. In the hydrogel, amoxicillin (AMOX) was chemically attached to GO via a bromelain enzyme-cleavable peptide linker (Leu-Leu-Gly) and released from the hydrogel in the presence of bacteria. Despite the high specificity and catalytic abilities of enzymes in controllable drug delivery, inactivation of bacterial secreted enzymes may result in the loss of enzyme responsiveness of such smart nanocarrier platforms. Coupling enzyme responsive NPs with other stimuli-responsive antibacterial agents to create synergistic antibacterial platforms may be a possible strategy for ensuring effective antibacterial activities. Bacterial infections also result in excess wound inflammation, leading to increased ROS levels in the microenvironment, which is a potential effective trigger for antibacterial agent delivery [160,161]. Based on this, various ROS-responsive nanocarrier systems were fabricated using ROS-sensitive substances, such as poly (propylene sulfide) (PPS), phenylboronic acid, poly(thioketal) (PK) and their derivatives [162,163,164,165]. Li et al. developed ROS-responsive mesoporous silica nanoparticles (MSNs) for controlled delivery of antibiotics in infected wounds, in which vancomycin was physically loaded into the amino-modified MSNs. Then, the ROS-sensitive thioketal linker grafted methoxy poly (ethylene glycol) (mPEG-TK) was further modified on MSN surfaces as a gatekeeper [164]. In vitro, the high H_2_O_2_ levels resulted in breakage of the TK linker and induced vancomycin release. The as-prepared van-mPEG-TK-MSNs successfully killed all the bacteria and cured the in vivo tissue infection after 14 days. Some metal-containing nanomaterials have been proven to have nanoenzyme activities, which can also be used as ROS-responsive antibacterial agents [166,167]. Tu et al. fabricated a novel multifunctional nanocomposite hydrogel wound dressing platform with synergistic performances of the antibacterial polymer and ROS-sensitive manganese dioxide (MnO_2_) nanosheets against bacterial proliferation (Figure 6C) [166]. In this work, the antibacterial hyperbranched poly-L-lysine (HBPL) was grafted on manganese dioxide (MnO_2_) nanosheet surfaces and the as-prepared HBPL-MnO_2_ were further used as crosslink agents to crosslink hydrophilic poly (PEGMA-co-GMA-co-AAm) to prepare the nanocomposite hydrogel. The MnO_2_ nanosheets are used as nanoenzymes to convert excess ROS on the MRSA-infected wound into oxygen and kill bacteria. The synergistic effects of HBPL and MnO_2_ nanoenzymes have been shown to kill up to 94.1–99.5% of *MRSA*, *E. coli* and *Pseudomonas aeruginosa* even at 10^9^ CFU/mL in vitro. In vivo, the MRSA-infected rat wounds were effectively treated by the hydrogel and the wound healing was significantly accelerated.

**Exogenous stimuli-responsive nanocarriers for antibacterial therapy.** Compared with bacterial microenvironment-responsive nanocarriers, stimuli-responsive nano-based drug delivery systems that can be induced by external stimuli, including light, magnet, and ultrasound among others possess unrivaled advantages in controlled release of antibacterial agents and infection treatment since these external stimuli can be easily acquired and precisely controlled without time and space limitations [168,169]. Among the external stimuli, because of its convenience, high-efficiency, non-invasion and the ability for spatiotemporal control, light is a particularly attractive tool that has been widely used in drug delivery, disease therapy and tissue engineering [170,171,172]. Lights, including ultraviolet (UV) light, visible light, and near-infrared (NIR) light have been investigated as triggers to combat bacterial infections by directly secreting antibacterial agents or killing bacteria through NIR-induced photothermal and photodynamic therapy [28,173]. Light-induced release of the preloaded drug from nanocarriers is one of the effective strategies for infection control [174,175,176]. Ballesteros et al. loaded AgNPs into a photosensitive nanogel and immobilized the nanogel on the surface of poly(ε-caprolactone) nanofibers mats [174]. Upon UV light irradiation (405 nm), the nanogel collapsed, resulting in responsive release of AgNPs into nanofiber mats and subsequent release of Ag^+^ to effectively kill bacteria. Studies have also reported on UV light-induced release of nitric oxide (NO) and visible light-induced release of carbon monoxide (CO) from nanocarriers to cure bacterial-infected wounds [175,176]. The limited penetration abilities of visible and UV light may impede their applications in deep skin tissues. Compared with visible and UV light, NIR light exhibits excellent tissue penetration abilities. Apart from the NIR-induced direct release of antibacterial drugs, the NIR-induced photothermal and photodynamic effects of nanomaterials are promising antimicrobial strategies. It has been well reported that nanomaterials including metal NPs (e.g., Au NPs [177] and Cu NPs [178]) metal sulfide/oxide nanomaterials (e.g., CuS [179], TiO_2_ [180] and MnO_2_ [181]), carbon-based nanomaterials (e.g., GO [182] and CNTs [183]), polymer NPs (e.g., polydopamine NPs [184]) and iron (III)-catechol complex (e.g., Fe^3+^-Tannic acid complex [185]) exhibit excellent photothermal responsive performances that can absorb NIR light and convert it into local heat to kill bacteria by destroying bacterial membranes and denaturing proteins as well as DNA. Photoactive hydrogels with attractive photothermal antibacterial capacities can be developed by combining the above-mentioned nanomaterials with hydrogels, and used as wound dressings to promote wound healing. Moorcroft et al. encapsulated gold nanorods and antimicrobial peptides (IK8) into thermal-cleavable liposomes and loaded the liposomes into a PEG hydrogel [186]. Given the photothermal effects of gold nanorods upon NIR irradiation at 860 nm, photothermal-induced release of IK8 and the intrinsic photothermal bactericidal capacity of gold nanorods exhibited excellent synergetic antibacterial effects against *P. aeruginosa* and *S. aureus*. In another study, a dopamine-coated Au nanorod (Au@PDA NR)-incorporated photothermal antibacterial hydrogel was prepared and further coated with *S. aureus* or *E. coli* pretreated macrophage membranes (Figure 7A) [187]. In vitro, due to the presence of surface-coated macrophage membranes, the as-prepared hydrogel specifically recognized and killed 98% of the bacteria via the superimposed photothermal effects of both Au nanorod and PDA within NIR irradiation for 5 min (Figure 7B). In vivo, the hydrogel exhibited positive therapeutic effects and promoted *S. aureus*-infected wound healing (Figure 7C). They crosslinked hyaluronic acid-grafted with tyramine (HT) and gelatin grafted with gallic acid (GGA) via a HRP/H_2_O_2_ catalytic system to form the injectable hydrogel, in which GO was loaded [188]. They crosslinked hyaluronic acid-grafted with tyramine (HT) and gelatin grafted with gallic acid (GGA) via a HRP/H_2_O_2_ catalytic system to form the injectable hydrogel, in which GO was loaded (Figure 7D). The hydrogel demonstrated a good photothermal antibacterial activity against *E. coli* and *S. aureus* and could significantly eliminate the infection of *E. coli* and promote the healing of *E. coli*-infected wound (Figure 7E). In addition, Fe^3+^-catechol complexes with good photothermal conversion properties have a great potential in developing NIR-responsive antibacterial hydrogel dressings [189,190,191]. In a typical study, dual dynamic-bond cross-linked hydrogels were prepared by crosslinking the protocatechualdehyde (PA)-Fe^3+^ complex with the quaternized chitosan (QCS) for treatment of infected wounds (Figure 7F). The inherent antibacterial activities of QCS and the excellent photothermal-antibacterial performances of the PA-Fe^3+^ complex endow the hydrogel with abilities to achieve high wound closure and promote MRSA-infected wound healing (Figure 7G,H) [191].

In a manner that is similar to photothermal therapy (PTT), photodynamic therapy (PDT) is an effective and minimally invasive strategy for combating bacterial infections by using some photosensitizers to secrete ROS under light irradiation and kill bacteria via ROS-induced oxidative stress of bacterial cells, and damage to intracellular components such as proteins, lipids and nucleic acids [21,192]. In recent decades, various inorganic and organic nano-based photosensitizers such as ZnO [193], TiO_2_ [194], CuO [195], MOFs [196,197], graphene [198], MXenes [199], black phosphorus (BP) [200], etc. have all n demonstrated excellent photothermal antibacterial activity. However, the low stability and poor water solubility of some photosensitizers inhibits their independent applications in infected wound sites. Taking advantage of hydrogels for wound healing, nano-photosensitizers loaded hydrogels with potential antibacterial and repair performances have shown inspiring applications in infected wound healing [28]. In 2019, Wang et al. incorporated Ag-doped TiO_2_ (Ag/TiO_2_) nanoparticles into PVA hydrogel to treat an infected wound (Figure 8A). The photodynamic activities of TiO_2_ NPs in visible wavelength range were greatly improved by the doping of Ag on TiO_2_ NP surfaces. Moreover, TiO_2_ NPs inhibited Ag^+^ release, thereby suppressing the potential toxicity of Ag^+^. Upon visible irradiation of 606 nm within 5 min, the hybrid hydrogel exhibited effective bacterial-killing abilities against both *E. coli* and *S. aureus* via light-induced production of ROS in the Ag/TiO_2_ system. Using this hybrid hydrogel as a wound dressing, it greatly inhibited the *S. aureus*-induced infection and significantly accelerated the wound healing [201]. In a similar study, due to production of singlet oxygen (^1^O_2_) of BP under visible light irradiation, black phosphorus nanosheets (BPs)-embedded chitosan hydrogels (BP@ CS) exhibited incomparable photodynamic bactericidal effects in vitro and in vivo (Figure 8B) [200]. Some porphyrin photosensitizers such as sinoporphyrin sodium (DVDMS) also have high singlet oxygen (^1^O_2_) yield and photodynamic antibacterial properties. However, under physiological conditions, DVDMS is not stable enough, which compromises its photodynamic activities. To overcome this challenge, Mai et al. incorporated nano-DVDMS into carboxymethyl chitosan (CMCS)-sodium alginate hydrogel (Figure 8C). In vitro, 10 μg/mL of the DVDMS -loaded hydrogel exhibited excellent antibacterial and anti-biofilm activities by eliminating almost 99.99% of *S. aureus* and *MRSA* under mild photoirradiation conditions (30 J/cm^2^, 5 min). The hydrogel has been proven to effectively inhibit bacterial growth while accelerating wound healing in burn infected models. Therefore, incorporating DVDMS into the hydrogel markedly improved the biological stability of DVDMS and enhanced the photodynamic antibacterial activities of DVDMS [202].

Despite mature applications of PTT and PDT on infection treatments, there are some concerns regarding the single use of PTT or PDT. For instance, to effectively alleviate bacterial growth, a high-temperature is required using PTT, which may damage normal cells and tissues around the wound [203,204]. The bactericidal effects of PDT are also dependent on elevated ROS production levels. However, excessive ROS may also induce inflammation, fibrosis, and necrosis of tissues during PDT treatment [205,206]. The heat can promote the transfer as well as ROS release by photosensitizers and increase bacterial membrane permeability to facilitate the entry of ROS into the bacterial cell and effectively kill the bacteria [207]. Moreover, a high-efficiency bacterial-killing capacity can be achieved by dual modes combination of lower temperature produced by PTT and moderate doses of ROS produced by PDT [208]. Therefore, the synergistic antibacterial therapy by combining PTT and PDT is a more effective and safer strategy than using single photothermal or photodynamic treatment [28,209]. In a recent study, a multifunctional black phosphorus (BP) hydrogel with antibacterial and antioxidant properties was developed for diabetic skin wound treatment [210]. In this study, 4-octyl itaconate (4OI) with antioxidant and anti-inflammatory properties was modified onto BP nanosheets and loaded into gelatin methacryloyl (GelMA) hydrogels (Figure 9A). In vitro, due to the photothermal effects of BP, a temperature of 52 °C was reached in the BP-based hydrogel under NIR irradiation for 5 min. This temperature has been proven to effectively kill bacteria and facilitate ^1^O_2_ permeation to enhance the PDT effects of BP. Benefiting from excellent PDT and PTT properties of BP, the as-prepared hydrogel exhibited synergistic bactericidal effects by killing 92.27% of *E. coli* and 96.69% of *S. aureus* upon 808 nm NIR irradiation for 5 min (Figure 9B) and effectively accelerated the closure of diabetic wound (Figure 9C). Similarly, Xie et al. prepared lignin-copper sulfide (LS-CuS) nanocomposites loaded PVA hydrogel, which exhibited good antibacterial and anti-biofilm capacities due to the synergistic PTT and PDT effects of CuS NPs under 808 nm laser irradiation [211]. Cui et al. fabricated a composite polyisocyanide (PIC)-based hydrogel with PDT and PTT dual antibacterial capacities in which cationic conjugated polythiophene (PMNT) was loaded and used as a PDT agent induced by white light. Water-insoluble conjugated polymer nanoparticles (CPNs-TAT) were also encapsulated and served as PTT agents triggered by NIR light (Figure 10). These complex hydrogels avoided PMNT aggregation and reduced the toxicity of CPNs-TAT, endowing the hydrogel with enhanced photodynamic and photothermal antibacterial activities [208]. Xiang et al. achieved synergistic antibacterial effects of PDT and PTT using carbon quantum dot decorated ZnO (C/ZnO) NPs and PDA, in which PDA was responsible for PTT under 808 nm NIR light and C/ZnO NPs was the source of ROS for PDT under 660 nm light. The sustained release of Zn^2+^ also enhanced the antibacterial properties. This system was shown to kill 99.9% of bacteria under dual-light irradiation and greatly promoted wound healing [212]. In a novel study, bacterial-targeting aggregation-induced emission (AIE) self-assembled nanoparticles (AIE NPs) with high water solubility, good biocompatibility, and good photostability were developed to achieve synergistic PTT and PDT effects against pathogens under 808 nm irradiation. Upon PTT/PDT treatment, the sterilizing efficiencies of AIE NPs were 99.9% and 99.8% for *S. aureus* and *E. coli*, respectively. In vivo, the AIE NPs-induced phototherapy efficiently eliminated *S. aureus* and promoted *S. aureus*-infected wound healing [213].

## 3. Intelligent Nanomaterials-Based Wound Dressings for Infection Detection and Treatment

Apart from antibacterial therapy, timely detection of bacteria at the wound site is of great importance for antibacterial wound dressings to provide appropriate and efficient anti-infection treatments. Currently, clinical detection of wound infection is usually performed by removing covered dressings, scraping wound exudates for further pathogen detection and by visual diagnosis based on some obvious clinical symptoms, including redness, swelling, pustule, and heat. The infections easily induce secondary wound damage [214,215]. Frequent changes of dressings may also induce secondary wound injury and increase pain in patients. Therefore, developing intelligent wound dressings that are capable of both detecting bacterial infections in-situ and providing timely antibacterial therapies for infection control is of great significance. Such wound dressings may overcome the disadvantages of traditional clinical approaches for infection detection and markedly improve the therapeutic outcomes as well as innovate management models for wound infections. Currently, intelligent wound dressings that simultaneously integrate wearable sensors and advanced drug delivery systems have achieved tremendous advances in wound infection diagnosis and treatment [215,216,217,218]. The unique optical and physicochemical properties of nanomaterials and rapid advances in nanotechnologies have endowed nanomaterials with great potentials in sensor and disease detection [219,220]. Advanced nanomaterials-based wound dressings that possess the functions of self-reporting of bacterial infections and antibacterial therapy have been widely investigated and applied in wound healing [221,222,223,224]. Zhou et al. developed bacteria-responsive smart wound dressings for simultaneous detection and inhibition of bacterial infections by simultaneously encapsulating self-quenched fluorescent dyes and antimicrobials into bacterial secreted toxins-lytic lipid vesicles and loading the vesicles into methacrylated gelatin (GelMA) hydrogels (Figure 11A). In vitro and in vivo, the as-prepared wound dressing provided a visual warning of *S. aureus-* and *P. aeruginosa*-associated infections by color changes of the dressing upon *S. aureus-* and *P. aeruginosa*-secreted toxin-induced release of fluorescent dyes and inhibited *MRSA* and *P. aeruginosa* proliferation by the released antimicrobial (Figure 11B–D) [221]. In 2020, Qiao et al. designed a Cyanine3 (Cy3) and Cyanine5 (Cy5)-co-modified silica nanoparticles (SNP-Cy3/Cy5) and co-encapsulated a UV-cleavable antibiotic prodrug (GS-LinkerMPEG) into PVA hydrogel to monitor bacterial infections and provide on-demand treatment. In this system, SNP-Cy3/Cy5 served as a pH-responsive fluorescent probe for detection of bacterial infections in which the fluorescence resonance energy transfer (FRET) effect between Cy3 and Cy5 could be broken in an acidic bacterial microenvironment and induce fluorescence changes of SNP-Cy3/Cy5. Moreover, up-conversion nanoparticles (UCNPs) were embedded into the hydrogels, which were used to transfer NIR light to UV light and induce the release of gentamicin from the UV-cleavable prodrug (GS-LinkerMPEG) to provide on-demand antibacterial treatment [223]. This nanoparticle composited hydrogel provided a new strategy for self-reporting and effective treatment of wound infections. In another study, a colorimetric nano-based wound dressing for point-of-care sensing and treatment of both drug-sensitive (DS) bacterial and drug-resistant (DR) bacterial infections was developed, which consisted of a pH indicator dye of bromothymol blue (BTB) and a chitosan coated nitrocefin and ampicillin-coloaded porphyrin-based metal-organic framework (MOF) PCN-224 (CP) (Figure 11E–H) [224]. In such a system, bromothymol blue (BTB) is used as an indicator for drug sensitive bacteria, the color of which changes from green to yellow under acidic environments of drug sensitive bacteria. Ampicillin-loaded CS-coated PCN-224 was hydrolyzed in acidic bacterial conditions to release antibiotic and kill the drug sensitive bacteria. In the presence of drug-resistant (DR) bacteria, nitrocefin would be released from PCN-224 and interacted with β-lactamase secreted by DR bacteria, and thus the dressing color would change to red to indicate DR bacterial infection. In such a situation, PCN-224 NPs would serve as PDT antibacterial agents to generate ROS under NIR irradiation and kill DR bacteria. Such a nano-based dressing system only takes 2–4 h to realize the sensing of bacterial infections using the naked eye with a low detection limit of 10^4^ CFU/mL for DR *E. coli*.

## 4. Conclusions and Future Perspectives

In this review, we elucidate on research progress of nanomaterials-based wound dressings in bacterial infection control and wound healing. Among the nanomaterials, metal and metal oxide nanoparticles such as AgNPs, AuNPs, ZnO NPs and bimetallic NPs that have intrinsic antibacterial properties are capable of efficiently killing bacteria and reducing bacterial drug-resistance through multiple antimicrobial mechanisms. Integrating these NPs into hydrogel-based wound dressings avoids the aggregation, prolongs the release and reduces the toxicity of NPs, which enhances the bacterial-killing performances of NPs and promotes infected wound healing. In addition, NPs, including polymer NPs, lipid NPs and inorganic NPs have been used as drug carriers to deliver and release antibacterial agents in a sustained manner. Given the specific physicochemical properties of nanomaterials and rapid nanotechnological developments, smart nanomaterials-based carriers that can respond to endogenous bacterial microenvironments such as acidic pH, specific toxins or enzymes secreted by bacteria and excess ROS, as well as external stimuli such as light to release antibacterial agents or trigger their antimicrobial properties in a controllable manner have also been widely applied in wound dressing and greatly improved antibacterial treatment outcomes. Nanomaterials with the ability to kill bacteria based on produced heat (PTT) or ROS (PDT) under light irradiation, exhibiting unparalleled advantages in infection treatment. Nano-based intelligent wound dressings with abilities for detecting bacterial infections in situ and providing timely antibacterial therapies will improve the management models for traditional wound infections and achieve better therapeutic outcomes.

Even if great progresses have been achieved in nanomaterial-based wound dressings for infected wound treatment, various challenges should be addressed before these dressings can be finally clinically applied. For example, the potential toxicity and long-term biosafety of nanomaterials should be improved. Greener synthesis of traditional nanomaterials using various bioresources or developing new nanomaterials from FDA-approved materials to yield more purified, nontoxic and even safely degradable nanomaterials can be the possible development direction of nanomaterials. Moreover, the existing smart nanomaterials with stimulus-responsive properties for infection detection and treatment require complex design and synthesis procedures, which increases the difficulty and costs for large-scale manufacturing. Developing such smart nanomaterials with simpler designs and reproducible processes by combining advanced nanotechnologies and manufacturing approaches may be a possible strategy for reducing costs and enabling clinical applications. Finally, infected wound healing is a complex issue, especially when multiple drug-resistant bacteria and biofilms occur at the wound site. New nanomaterials-based wound dressings with abilities for targeting different bacterial pathogens and personalized combinational therapies to ensure effective antibacterial treatments should be further developed. Moreover, some growth factors, genes or stem cells that greatly accelerate the wound healing can be incorporated into nano-based wound dressings and co-delivered with antibacterial agents. For such wound dressings, rational and orderly delivery of antimicrobials and repair promoting factors require careful attention. In summary, nano-based wound dressings are novel and efficient platforms for infected wound treatment. We hope that more collaborative efforts of researchers from different research fields will be devoted to development of new generations of nano-based wound dressings to solve the above challenges.

## Figures and Tables

**Figure 1 antibiotics-12-00351-f001:**
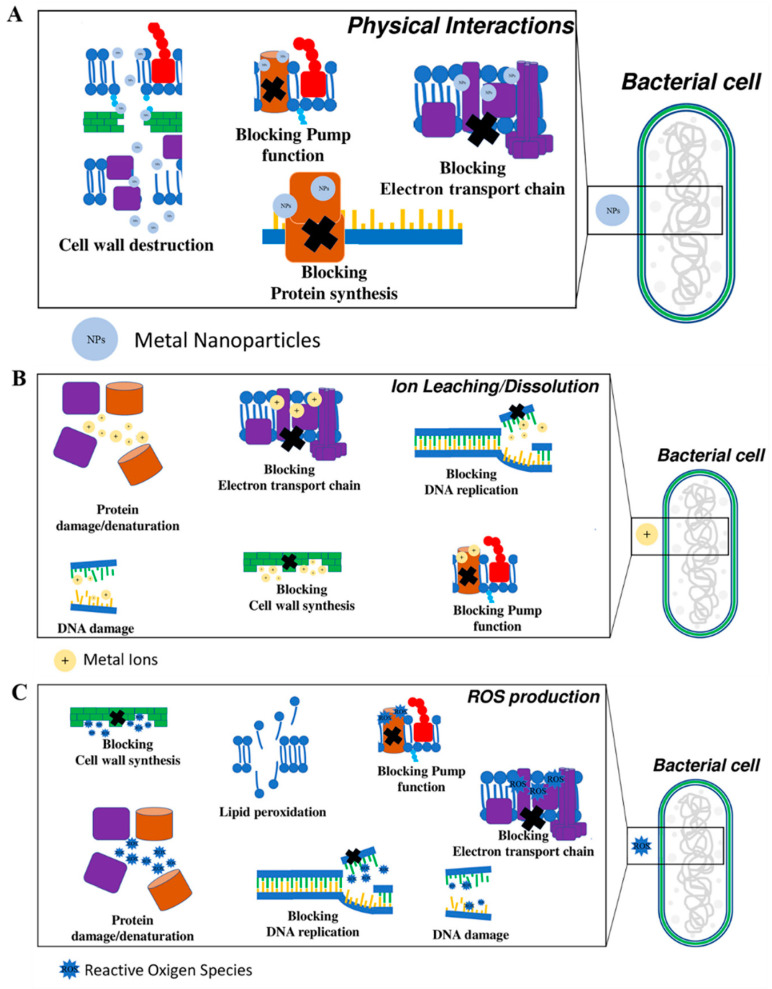
Schematic description of the antibacterial mechanisms of metal and metal oxide nanoparticles. (**A**) Antibacterial mechanisms due to a physical interaction mediated by nanoparticles (NPs). (**B**) Antibacterial mechanisms due to the release of metal ions (^+^) by NPs. (**C**) Antibacterial mechanisms due to the ROS produced by NPs. Reproduced with permission from ref [41], Copyright 2022 MDPI.

**Figure 2 antibiotics-12-00351-f002:**
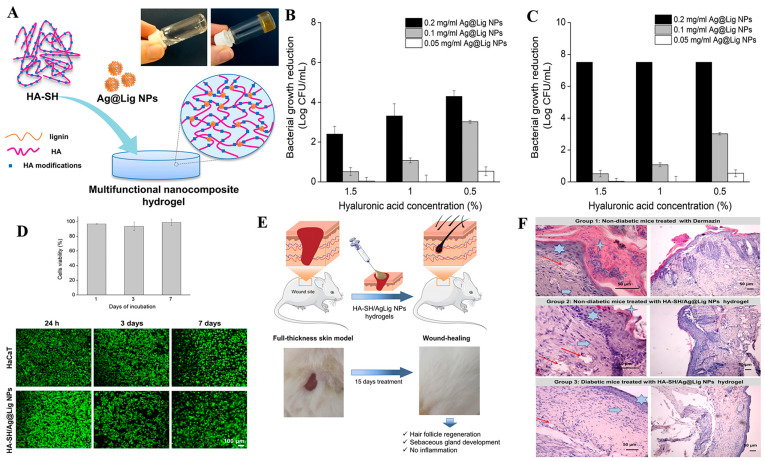
(**A**) Schematic representation of the preparation process and possible self-assembly mechanism of Ag@Ligin NPs-loaded hyaluronic acid (HA) nanocomposite hydrogel. Profile of the hydrogel’s antibacterial activities against (**B**) *Staphylococcus aureus* and (**C**) *Pseudomonas aeruginosa*. (**D**) Viability of keratinocytes after 1, 3 and 7 days coculture with the Ag@Lig NPs-loaded hydrogel. (**E**) Representative images of full-thickness skin wounds in a mouse model before and after application of the nano-enabled hydrogels. (**F**) Hematoxylin-eosin-stained sections of wound tissues across different treatments. Reproduced with permission from ref [59], Copyright 2021 Elsevier.

**Figure 3 antibiotics-12-00351-f003:**
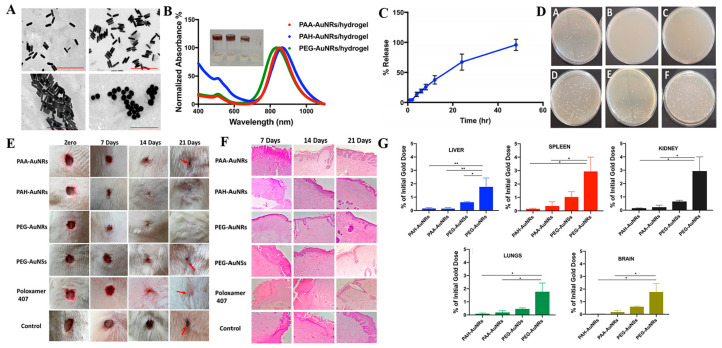
(**A**) TEM images of AuNPs showing different sizes and surface modifications. (**B**) UV-vis absorption spectra and photos of AuNRs-hydrogels with different surface modifications. (**C**) In vitro release curve of AuNRs from the PEG-AuNRs loaded poloxamer 407 hydrogel. (**D**) Overnight bacterial cultures of wound-contacted swaps after 14 days of wound treatment by different hydrogels. (**E**) Photographs and (**F**) H&E histological examination of rat wounds after treatment by hydrogels of PAA-AuNRs, PAH-AuNRs, PEG-AuNRs, PEG-AuNSs, poloxamer 407 hydrogel and control for different time. (**G**) Deposition percentages of AuNPs into different rat’s organ. Reproduced with permission from ref [69], Copyright 2019 Elsevier.

**Figure 4 antibiotics-12-00351-f004:**
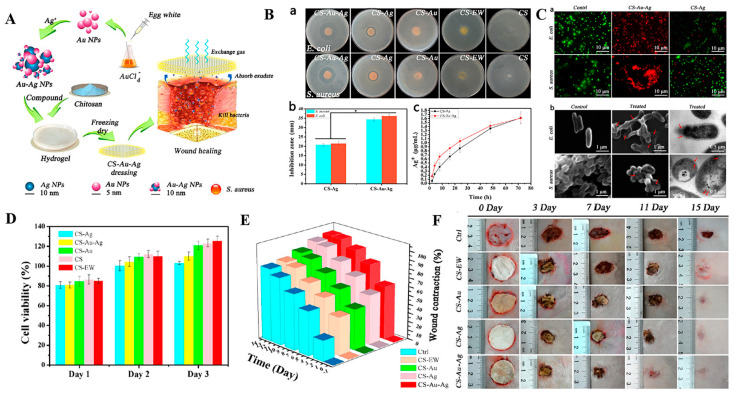
(**A**) Conceptual illustration of the preparation process for bimetallic Au-Ag NPs loaded chitosan antibacterial hydrogel wound dressing. (**B**) Antibacterial activities of different hydrogel samples against *E. coli* and *S. aureus* and Ag^+^ release kinetics of CS-Ag and CS-Au-Ag hydrogel. (**C**) Representative fluorescent images and SEM and TEM micrographs of *S. aureus* and *E. coli* treated with CS-Au-Ag and CS-Ag hydrogel. (**D**) Viability of cells incubated with different hydrogel samples. (**E**) Wound contraction ratio and (**F**) macrographs of wounds covered with different hydrogel samples at days 0, 3, 7, 11, and 15. Reproduced with permission from ref [79], Copyright 2017 American Chemical Society.

**Figure 5 antibiotics-12-00351-f005:**
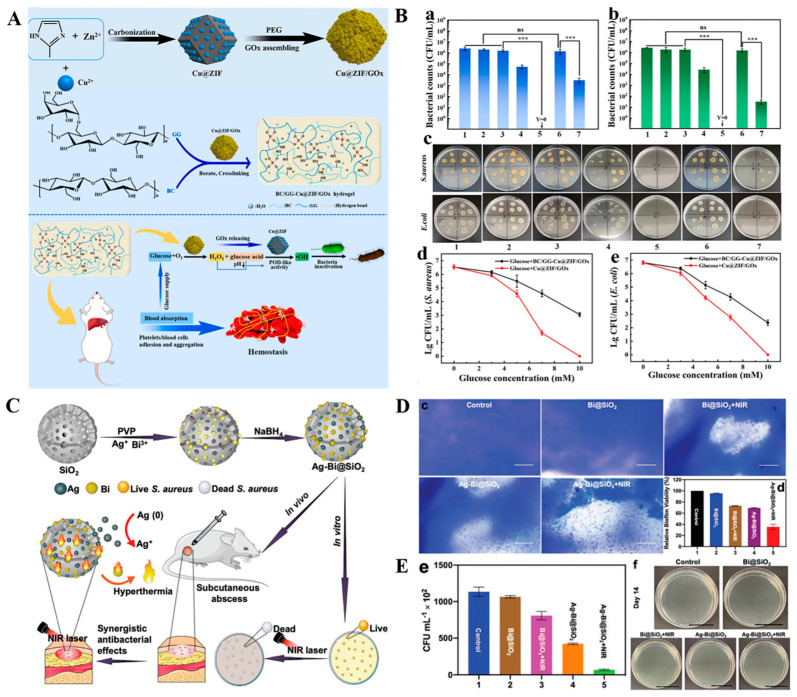
Inorganic nanocarriers for antibacterial therapy. (**A**) Schematic illustration of the preparation procedure of BC/GG-Cu@ZIF/GOx hydrogel and its catalytic cascaded reaction for bacteria inactivation. (**B**) In vitro antibacterial efficacy of various samples against (**a**) *S. aureus* and (**b**) *E. coli* after 6 h incubation; (**c**) Representative photographs of bacterial colonies of each set. 1. PBS, 2. Glucose, 3. Cu@ZIF/GOx + PBS, 4. GOx + glucose, 5. Cu@ZIF/GOx + glucose, 6. BC/GG-Cu@ZIF/GOx + PBS, 7. BC/GG-Cu@ZIF/GOx +glucose; (**d**,**e**) Antibacterial efficacy of Cu@ZIF/GOx treated with different concentrations of glucose after 6 h incubation. Reproduced with permission from ref [131], Copyright 2022 Elsevier. (**C**) Schematic description for the preparation of Ag-Bi@SiO_2_ NPs and its synergistic antibacterial effects. (**D**) In vitro anti-biofilms analysis of Ag-Bi@SiO_2_ NPs with different treatments by (**c**) staining the biofilms with crystal violet and (**d**) quantification of the biofilm activity using XTT assay. (**E**) Determination of the in vivo anti-infection efficacy by (**e**) based on the numbers of survived MRSA in infected skin after 14 d of therapy and (**f**) spreading the bacteria on plate and taking corresponding photographs. Reproduced with permission from ref [135], Copyright 2020 Wiley-VCH.

**Figure 6 antibiotics-12-00351-f006:**
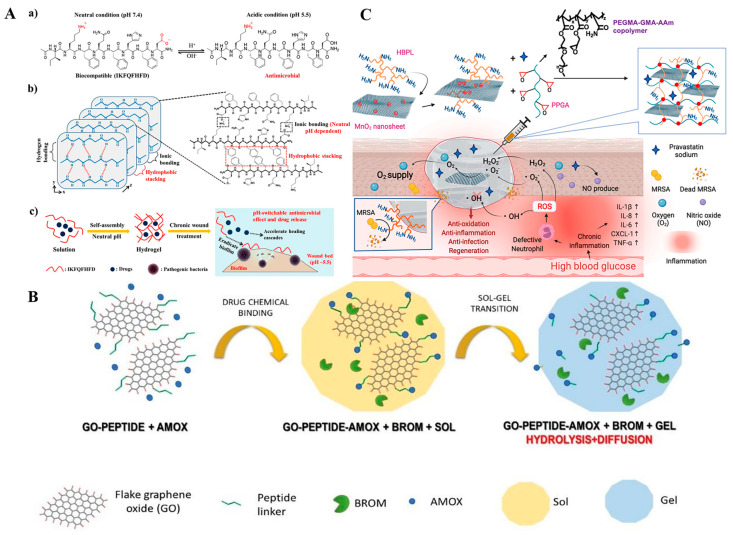
Endogenous stimuli-responsive nanocomposite hydrogel wound dressings for antibacterial therapy. (**A**) Schematic illustration of pH-switchable antimicrobial peptide (IKFQFHFD)-based nanofiber hydrogel wound dressing. (**a**) Chemical structure of IKFQFHFD under neutral and acidic conditions; (**b**) Proposed molecular arrangement of IKFQFHFD to form self-assembled supramolecular nanofiber networks of hydrogel at neutral pH; (**c**) Conceptual illustration of IKFQFHFD during the formation of hydrogel at neutral pH and response to the released peptides for biofilm clearance at acidic conditions to promote healing of chronic wound healing. Reproduced with permission from ref [152], Copyright 2019 American Chemical Society. (**B**) Schematic of an enzyme-responsive nanocomposite hydrogel for the controllable release of antibiotic to treat bacterial infection, in which amoxicillin (AMOX) was chemically attached to graphene oxide (GO) through a bromelain enzyme-cleavable peptide linker (Leu-Leu-Gly) and then loaded into sodium alginate hydrogel. Reproduced with permission from ref [159], Copyright 2021 MDPI. (**C**) A ROS-sensitive hyperbranched poly-lysine (HBPL) modified manganese dioxide (MnO_2_) nanosheets composite hydrogel for antibacterial treatment of MRSA-infected diabetic skin wound, in which MnO_2_ nanosheets were used as nanoenzymes to convert the excessive ROS production by bacterial infection into oxygen and synergistically kill bacteria with HBPL. Reproduced with permission from ref [166], Copyright 2022 Elsevier.

**Figure 7 antibiotics-12-00351-f007:**
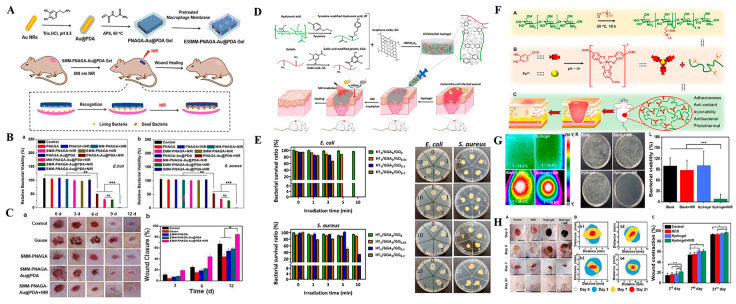
Nanocomposite hydrogel wound dressings with efficient photothermal bactericidal performances for the treatment of infected wound. (**A**) Schematic illustration of a dopamine-coated Au nanorod (Au@PDA NR)-incorporated nanocomposite hydrogel with a targeted binding to specific bacteria by coating with *S. aureus* or *E. coli* pretreated macrophage membrane and efficient photothermal antibacterial activity for promoting wound healing. (**B**) In vitro antibacterial activity of different samples and (**C**) in vivo wound healing activity of *S. aureus* infected wound on the dorsal area of rats after the treatment of hydrogels under NIR irradiation. Reproduced with permission from ref [187], Copyright 2020 Elsevier. (**D**) Preparation and therapeutic mechanism of graphene oxide (GO) composited hyaluronic acid (HT)/gallic acid-grafted gelatin (GGA) hydrogel for *E. coli* infected wound healing. (**E**) In vitro photothermal bactericidal activities of the as-prepare hydrogel against *E. coli* and *S. aureus* after NIR irradiation for 0, 1, 3, 5, and 10 min. Reproduced with permission from ref [188], Copyright 2022 Elsevier. (**F**) A photoactive antibacterial hydrogel wound dressing for the treatment of infected wound by crosslinking protocatechualdehyde (PA)-Fe^3+^ complex with the quaternized chitosan (QCS). (**G**) In vitro photo-triggered antibacterial property and (**H**) in vivo MRSA-infected wound healing evaluation of the hydrogel. Reproduced with permission from ref [191], Copyright 2021 American Chemical Society.

**Figure 8 antibiotics-12-00351-f008:**
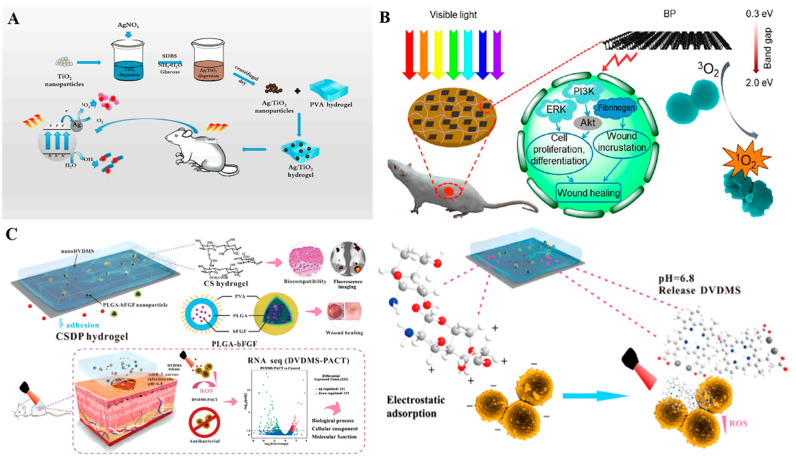
Nanocomposite hydrogel wound dressings with efficient photodynamic bactericidal performance in the treatment of infected wound. (**A**) The schematic illustration of the synthesis route of Ag/TiO_2_-incorporated polyvinyl alcohol hydrogel and its application in the prevention of bacterial infection during wound healing through the photodynamic antibacterial activity of TiO_2_ upon visible irradiation at 606 nm. Reproduced with permission from ref [201], Copyright 2019 Elsevier. (**B**) A black phosphorus nanosheets (BPs)-embedded chitosan hybrid hydrogel (BP@CS) for visible light-triggered photodynamic therapy to accelerate bacteria-accompanied wound healing by BP-induced singlet oxygen (^1^O_2_) production and activation of signaling pathways. Reprinted from ref [200]. Copyright 2018 American Chemical Society. (**C**) Schematic illustration of a photosensitizer (DVDMS) loaded multifunctional hydrogel for synergizes bacterial clearance and skin regeneration of the infected burn wounds. Reprinted from ref [202]. Copyright 2020 American Chemical Society.

**Figure 9 antibiotics-12-00351-f009:**
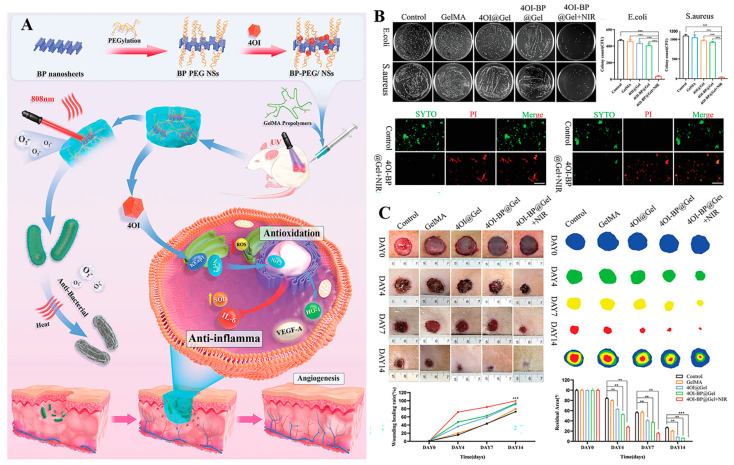
(**A**) Schematic illustration of the preparation procedure of the 4-octyl itaconate (4OI)-modified black phosphorus (BP) nanosheets loaded gelatin methacrylamide hydrogel to produce a new photothermal therapy (PTT) and photodynamic therapy (PDT) system under single NIR irradiation (808 nm) with antibacterial properties for diabetic wound healing. (**B**) In vitro antibacterial effects of various hydrogels against *E. coli* and *S. aureus* with or without NIR irradiation. (**C**) In vivo wound closure rate by the hydrogel during the healing process of diabetic infected wound. Reprinted from ref [210]. Copyright 2022 Wiley-VCH.

**Figure 10 antibiotics-12-00351-f010:**
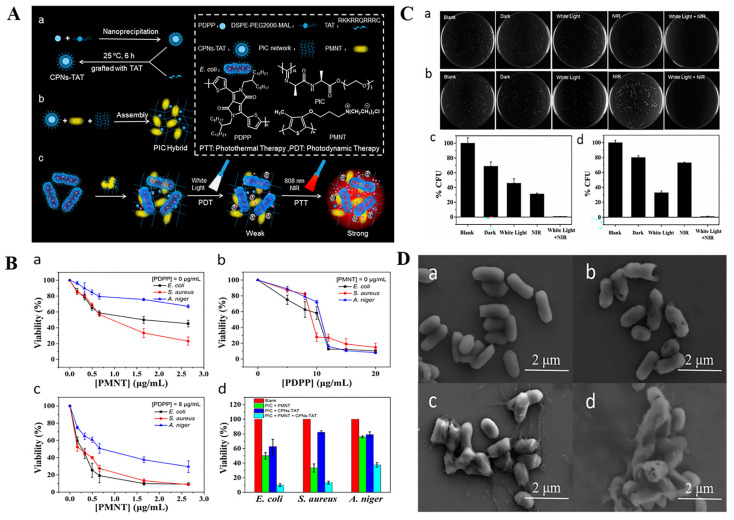
(**A**) Schematic illustration of the polyisocyanide (PIC)-based hybrid hydrogel for synergistic antibacterial application of photodynamic therapy (PDT) and photothermal therapy (PTT) under both white and NIR light irradiation. (**B**) In vitro growth inhibition of various hybrid hydrogel against *E. coli, S. aureus*, and *A. niger* (**a**) under white light irradiation; (**b**) under NIR light irradiation; (**c**) under white light and NIR light irradiation; and (**d**) histogram of the inhibition effects. (**C**) In vitro antibacterial effect of the PIC hybrid hydrogel against *E. coli* and *S. aureus* under different light sources using standard plate counting method. (**D**) SEM images of *E. coli* after different treatments: (**a**) incubated with the PIC hydrogel without any light irradiation; (**b**) incubated with the PIC/CPNs-TAT complex without any light irradiation; (**c**) incubated with the PIC/PMNT complex with white light irradiation; (**d**) incubated with the PIC hybrid with white light and NIR light irradiation. Reprinted from ref [208]. Copyright 2020 American Chemical Society.

**Figure 11 antibiotics-12-00351-f011:**
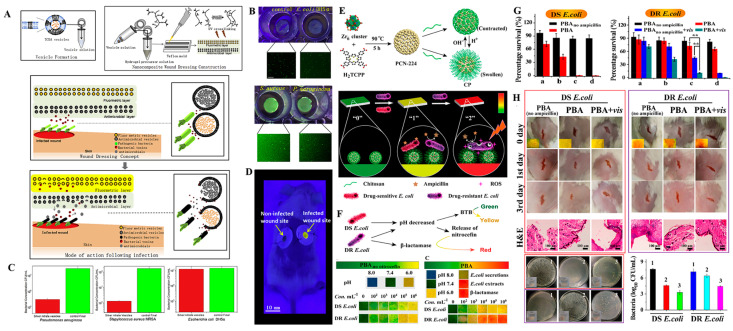
Nano-based intelligent wound dressings for the all-in-one infection detection and treatment. (**A**) The schematic illustration of bacteria-responsive vesicles loaded with hydrogel wound dressing for fluorescent detection of pathogenic bacteria and antibacterial treatment, in which the self-quenched fluorescent dyes and antimicrobials are encapsulated into the bacteria-cleavable vesicles. (**B**) In vitro fluorescence response of the as-prepared wound dressing when cocultured with different bacteria. (**C**) In vitro antimicrobial behaviors of the as-prepared wound dressing against three bacterial strains. (**D**) Representative images of showing the colorimetric sensing property of the wound dressing when applied to the infected wounds. Reprinted from ref [221]. Copyright 2018 Elsevier. (**E**) The schematic illustration of the colorimetric nano-based wound dressing for point-of-care sensing and treatment of both drug-sensitive (DS) bacteria and drug-resistant (DR) bacteria based on a pH indicator dye of bromothymol blue (BTB) and a chitosan coated nitrocefin and ampicillin-coloaded porphyrin-based metal-organic framework (MOF) PCN-224 (CP). (**F**) The schematic diagram showing the wound dressing in sensing bacterial infection and drug resistance. (**G**) Viability of drug-sensitive (DR) *E. coli* and drug-resistant (DR) *E. coli* incubated on the hydrogel with or without light irradiation. (**H**) In vivo disinfection effects of the intelligent wound dressing in mice wound model. Reprinted from ref [224]. Copyright 2019 American Chemical Society.

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
