# Peer review of "Nanomaterials-Based Wound Dressing for Advanced Management of Infected Wound"

_antibiotics, 2023, doi:10.3390/antibiotics12020351_

Round 1
Reviewer 1 Report
This review focuses on the use of antibacterial nanomaterial-based wound dressings for the treatment of chronic wounds. The manuscript is well-written; however, revisions are required. The manuscript should be revised in response to the comments.
- The authors should seek editing assistance from someone with full professional proficiency in English.
- On pages 1 and 2, lines 34-37, the authors should go into more detail about the wound healing process. The key events of each phase should be explained briefly. (For the reference: https://www.sciencedirect.com/science/article/abs/pii/S0141813021021541)
- Metal and metal oxide nanoparticles could be easily incorporated into electrospun nanofibers. The authors should consider including some studies on such antibacterial electrospun nanocomposite scaffolds in the treatment of chronic wounds.
- It would be helpful if the authors included a figure in subsection "2.1 Metal and metal oxide nanoparticles as intrinsic antibacterial agents for antibacterial treatment" depicting the antibacterial mechanisms of metal-based nanoparticles.
- On page 12, lines 243-245, polycaprolactone (PCL) and poly(lactic-co-glycolic acid) (PLGA) are not natural polymers. The sentence should be revised.
- Niosomes are promising lipid-based nanocarriers for drug delivery. The authors should discuss the role of these carriers in the delivery of antibiotics for wound healing applications. (For the reference: https://onlinelibrary.wiley.com/doi/full/10.1002/jbm.b.35039)
- Doped bioactive glass nanoparticles (e.g., zinc ion-doped bioactive glass nanoparticles and silver-strontium-doped mesoporous bioactive glass nanoparticles) have been used for wound healing due to their excellent antibacterial properties. It would be interesting if the authors discussed the use of doped bioactive glass nanoparticles in wound healing in light of recent published articles on the subject.
- All "et al.," "in vitro," "in vivo," and "in situ," as well as all scientific names, should be written in italics throughout the manuscript.
Reviewer 2 Report
The review is well organized and treats very deeply and clearly the proposed topic. I would like to accept it as it is.
Author Response
We appreciate very much for the valuable comments to this manuscript.
Reviewer 3 Report
Dear Authors,
The review article looks good and have shown extensive work nanomaterial based wound dressing. I would like to accept the article.
Author Response

(The authors gave the same response as above.)
